# "Putin's War of Choice": U.S. Propaganda and the Russia–Ukraine Invasion

**Aaron Hyzen * and Hilde Van den Bulck ***

Department of Communication, College of Arts and Sciences, Drexel University, Philadelphia, PA 19104, USA
* Correspondence: ah3739@drexel.edu (A.H.); hdv26@drexel.edu (H.V.d.B.)

**Abstract:** The Russian invasion of Ukraine on 24 February 2022 ignited propaganda efforts from the U.S. executive branch of government and the U.S. media, as the country tried to position itself towards the war not just in the eyes of its citizens but of the entire world as part of its geopolitical power position. A comparative quantitative and qualitative analysis of official U.S. communications and U.S. partisan media coverage in the first week of the invasion aims to uncover how the U.S. government set the agenda and framed the events, and to what extent the media copied or diverged from this agenda-setting and framing. The results suggest a narrow focus and distinct framing on the part of the U.S. government, partly taken over by partisan media. The latter also touched on other topics that fit media logic and provided some counter-frames in line with their ideological positions, yet overall confirmed the dominant framing of the war as unjust, unprovoked and premeditated, as Putin's choice, and the position of the U.S. as the leader of the free world and defender of democracy.

**Keywords:** propaganda; war propaganda; government propaganda; agenda-setting; framing; Ukraine; Russia; U.S.A.

## 1. Introduction

"American people are standing with the people of Ukraine as they suffer an unjustified, unprovoked, and premeditated attack by Russia's military forces" (SD 2022a, 2/25, p. 2). When Russia invaded Ukraine on 24 February 2022, it ignited reactions around the globe. The U.S. had to position itself towards the war not just in the eyes of its citizens but of the entire world as part of its geopolitical power position. This study provides a comparative quantitative and qualitative analysis of official communication from the executive branch of the U.S. government and U.S. partisan media coverage in the first week of the war as a propaganda campaign, aimed at setting the agenda and framing the position of the U.S. towards the events with an eye to affect public opinion.

A powerful means to tailor and control public opinion, propaganda is rarely more urgent and important than in times of war, for the countries having an armed conflict and for their allies and enemies. Propaganda has long been recognized as a forceful war weapon. For example, in the early 1930s, Poland proposed to the League of Nations to enact "moral disarmament", specifically propaganda, seen to incite public opinion to aggression. The Soviet Union similarly proposed to ban "ideological aggression" (Murty 1968, pp. 233–34). Currently, the U.N.'s International Covenant on Civil and Political Rights, a key human rights treaty, states in Article 20, "Any propaganda for war shall be prohibited by law".

Regardless, war times are rife with propaganda as governments and media engage in their own and are wary of their enemies' propaganda campaigns. At the outset of the Russian–Ukraine war, in early 2022 March, Russian media outlet RT News was de-platformed in many parts of the world to avoid Russian propaganda "polluting" minds or contradicting home propaganda (Roettgers 2022). Following suit, Russia banned and cut access to Western news media, including CNN, CBS, DW and BBC News (Sherwood and Milmo 2022).

War exacerbates the complex relationships between governments and news media that all express ideological goals and commitments. Governments recognize the importance of propaganda, invest vast resources in communication and need the media to get their messages out. The news media, in turn, rely heavily on the government for information, sources and access to war zones but have their own political agenda and information goals, potentially creating tension between truth and loyalties. One year into the Russian–Ukraine conflict, The Guardian (Koshiw 2023, par. 1) reported the following:

> After avoiding criticism of the authorities at the start of the war, Ukrainian journalists have begun reporting allegations of corruption by officials again. But wartime censorship and the army's role in protecting their country from an existential threat has made reporting on the military a challenge.

The U.S. is related to the conflict as the self-proclaimed "leader of the free world" and with many political and economic interests at stake. Since the 2014 Russian annexation of Crimea, some identify the Russia–Ukraine conflict as a proxy war between "Russia and the West (the United States, NATO, and member states of the European Union)" (Hughes 2014, p. 106; Foster 2022). Therefore, the U.S. required propaganda campaigns to declare its position towards, evaluation of and actions relating to the war as a message to its citizens, to the warring parties and to its allies. As such, it provides an interesting case to understand how propaganda operates in the 21st Century.

Scholars have identified propaganda dissemination as far back as 300 B.C. (Murty 1968) and up to the upsurge of media and communication networks in the 20th and 21st centuries (Jowett and O'Donnell 2019). Recent propaganda studies (Bakir et al. 2019; Howard and Kollanyi 2016; Wardle and Derakhshan 2018; Woolley and Howard 2016, 2017) focus on developments in big data, algorithms and wider AI within digital networks. The affordances of digital technologies have increased the capabilities of information warfare (Di Pietro et al. 2021), political communication, computational propaganda (Woolley and Howard 2016; Woolley and Howard 2017) and propaganda through social media (Wanless and Berk 2017). They allow rapid, cheap and endless repetition of propaganda across media platforms (Hyzen 2023a). Nevertheless, legacy media, including television, remain consequential in propaganda dissemination and particularly in war propaganda, the case we analyze here.

Knightley's (2004) seminal *The First Casualty* argues that the U.S. government has become increasingly successful in war propaganda and in managing the media, especially since the 1993 Gulf War, as a wealth of literature illustrates (Hiebert 2003; Boyd-Barrett 2004; De Franco 2012; MacArthur 2004). Herman and Chomsky's (1988) propaganda model predicts collusion between government and mass media to filter out information unfavorable to elite interests. Recent scholarship has updated the model to account for the contemporary media ecology and for additional filters (Boyd-Barrett 2017; Pedro-Caranana et al. 2018; Klaehn et al. 2018; Hyzen 2023b). Specifically, we argue that "Legal Standing", i.e., legality vs. illegality, acts as an ideological frame for propaganda purposes, especially in times of war (Hyzen 2023b).

Combining propaganda studies with insights into relationships between journalists/media and elite (government) sources, we argue that contemporary examples of wartime propaganda represent a form of immediate propaganda that originates from the government and media that rely on the indexing power (Bennett 1990) of their interactions to set the agenda of what aspects of the war are talked about and to influence how they are talked about. We show how the case fits a conceptualization of propaganda as a tangible expression of ideology aimed at enforcing ideological goals, managing opinion and consolidating loyalties. Specifically, how propaganda is not necessarily limited to mis/disinformation but includes factual information and can emphasize the truth if that best serves the overall strategy (Hyzen 2021). The opening quote holds a number of truths: many Americans stand with Ukraine, the war arguably does not follow the rules of a just war, and the invasion must have been planned. Yet, the message is highly charged,

ideological propaganda, for instance in stating the war was entirely unprovoked, as we will discuss.

After this introduction, we provide a theoretical exploration of these issues, resulting in three broad research questions. Next, we explore the questions empirically for the first week of the war (2/24-3/3/2022), when propaganda is expected to reflect the informational and ideological priorities of interested parties, i.e., the U.S. government and news media, to establish premises, narratives and perceptions, i.e., frame the conflict moving forward. We combine a quantitative analysis of the war-related topics covered (agenda-setting) and a qualitative analysis of the framing of these topics in official press briefings of the White House and State Department and in a sample of war coverage in partisan U.S. cable news programs, *CNN newsroom* and *Fox News Special Report*. The paper concludes with reflections on how this case fits current understandings of propaganda and state–media relationships.

## 2. Theoretical Framework

### 2.1. Propaganda and War

Propaganda is a means of communication to spread ideas and achieve ideological goals. As such, it is an expression of power (Hyzen 2021, p. 3483). Here, ideology is conceived of in two parts: Martin's (2015) concise program of values + beliefs = opinion and Williams (1975) definition as a coherent, assumed or naturalized set of interlocking beliefs, ideas and concepts. Thus, propaganda is a method to spread ideas, as well as enforce and coherently stabilize ideological views and attitudes to strategically manage opinion and to secure "loyalties" (Price 1994). For example, establishing that the war was "unprovoked" rests upon securing the premise of when the war began, whether in February 2022 or whether earlier events warrant inclusion in the conflict. U.S. officials and media carefully select or eliminate prior events to support ideological views cogent to their propaganda goal. Some scholars subscribe to propaganda's negative connotations (Benkler et al. 2018, p. 28), while we take a neutral position towards propaganda as an object of study and communication strategy (Hyzen 2021; Bernays 1928/2005).

Contemporary literature considers propaganda a subset of strategic communication (Hallahan et al. 2007; Zerfass et al. 2018). On the international (geo-)political stage, Murty (1968) characterizes propaganda campaigns between states as "ideological aggression", an aspect of warfare. Similarly, propaganda has been identified as a mode of psychological warfare or information warfare, to establish "information dominance" (Libicki 1995, p. 88), a soft power tactic to achieve political goals alongside hard power military operations (Nye 2004). As such, it is an important tool in proxy wars, i.e., as in the case of the U.S. involvement in the Russia–Ukraine conflict (Boyd-Barrett 2021).

Government or state propaganda is never more "urgent" and acknowledged than in times of war. Lasswell's *Propaganda Technique in the World* War (1927/2013) provides a road map for war propaganda operations to serve broad strategic aims, which remain astute and relevant to our case. Lasswell argued the three chief components of war are military pressure, economic pressure and propaganda (p. 9). He identifies four main strategic objectives of propaganda in times of war: to mobilize hatred against the enemy; preserve the friendship of the allies; preserve the friendship and, if possible, procure the co-operation of neutrals; and demoralize the enemy (p. 195). To these aims, propaganda has dual use, messages aimed at demoralization and destabilization of public support of an enemy also serve the purpose of domestic mobilization and support for warfare (Bolin et al. 2016). These principles, likewise, apply to countries involved in proxy wars (Boyd-Barrett 2021). We argue, furthermore, that such propaganda reaches both intended and unintended audiences with variable consequences (Hyzen 2023a).

Bernays sharpened Lasswell's program to establish six principles of strategy for war propaganda:

1.  Fasten the war guilt on the enemy.
2.  Claim unity and victory, in the name of history and deity.
3.  State war aims [. . .] Security, peace, a better social order, international laws [. . .].

4.  Strengthen the belief of the people that the enemy is responsible for the war, with examples of the enemy's depravity.
5.  Make the public believe that unfavorable news is really enemy lies. This will prevent disunity and defeatism.
6.  Follow this with horror stories [which] should be made to sound authoritative. (Bernays 1942, p. 236)

Crucially, these strategic goals cannot be achieved only through mis/disinformation, part of the broad category of "fake news" that propaganda tends to get lumped in with (e.g., Wardle 2018; Wardle and Derakhshan 2018). Though mis/disinformation will likely be deployed in the process of a propaganda campaign, facts and truths play an equal if not more important role (Hyzen 2023a). Official spokespersons do not simply produce outright lies as they are precarious—if revealed they can damage the propagandist's goals—but carefully craft information into strategic narratives (Schmitt 2018). The important work of propaganda is to place truths within distorted and manipulated contexts, to repetitively emphasize favorable facts, while downplaying unfavorable information to achieve the desired ideological goal. As such, it is important to analyze the veracity of propaganda claims and premises. This positions a propaganda message vis à vis misinformation, i.e., mistakenly convened inaccurate information, and disinformation, i.e., purposely convened inaccurate information. Importantly, propaganda serves to bring the target towards the truth or away from the truth in more nuanced ways (Hyzen 2023a). Of course, "the truth" is a larger, more complex construction than facts or true information, as it can involve both omissions and interpretationsr.

These reflections result in the broad research question (RQ1): how does the U.S. government propagate the Russia–Ukraine war and its involvement in it? (What propaganda strategies can be identified?)

## 2.2. State Propaganda and the Media

News media remain invaluable for governments to disseminate and distribute their propaganda, especially in times of war, with both the state and media aiming to "set the agenda". Agenda-setting (McCombs and Shaw 1972) refers to how the media choose news items: particular issues are emphasized whereas others are marginalized, choices that influence public opinion (De Franco 2012, p. 10). However, governments, too, attempt to affect news prevalence, drawing attention to or deflecting from certain aspects of a war and/or a country's involvement. Lippmann (1957) reflects that true propaganda required "some form of censorship [...] access to the real environment must be limited, before anyone can create a pseudo-environment" (p. 43). Likewise, Herman and Chomsky (1988) argued that information is omitted from mainstream media through pressure from power interests to support propaganda campaigns. As such, wartime agenda-setting by governments and media is as much deflecting attention as manipulating the prevalence of information.

Understanding the relationship between the executive branch and the fourth estate of the media, meant to hold democratic government accountable, has been the subject of much research. Bennett (2004) identifies four factors that affect these relationships: a journalist's personal and professional values, organizational routines, economic (business model) constraints, and information and communication technology characteristics. Sourcing routines especially have received considerable scholarly attention (e.g., Tuchman 1978; Gans 1979; Broersma et al. 2013; Berkowitz 2019), showing how the need for credibility results in a strong reliance on elite sources, including political elites and other state and government officials (Schudson 2002). Journalists' and the media's ideological positions, business considerations and time–space restraints affect this reliance on elite sources, next to the particular news topic. Whether these media–elite source relationships are a matter of "dueling, dancing or dominating" (Carlson 2009, p. 526) remains a topic of debate. However, empirical research confirms Bennett's (1990) notion of indexing, i.e., U.S. news media's tendency to follow elite sources' take on an issue, especially when there is relative consensus amongst political elites. These relationships are of particular relevance in times of

crisis, such as war (Mermin 1999; McChesney 2002; Kennis 2009), when media can become less critical, particularly in the early days of a military conflict, the focus of our study.

Official press briefings from executive government branches constitute crucial spaces for journalists to obtain views on an issue from elite political sources and, thus, powerful moments for governments to obtain the desirable ideological positionings in the media, whose coverage then helps set the agenda. This space is the media's domain and is a capacity the government seeks to utilize to disseminate its propaganda. Though the U.S. is not a direct military participant in the Russia–Ukraine war, previous studies have shown the U.S. has treated the longer-standing Russia–Ukraine conflicts as a proxy war in which it has vested interest, funding pro-Ukrainian propaganda campaigns following the Russian annexation of Crimea (Bolin et al. 2016). Since the 2014 invasion, the U.S. has closely coordinated propaganda narratives and communications with the Ukrainian government and with respective news media (Boyd-Barrett 2017). The current conflict is expected to show a continuation of such operations.

These considerations lead to the research question (RQ2): to what extent do the U.S. media pick up U.S. government propaganda attempts to set the agenda?

### 2.3. War Propaganda: Strategies and Tactics

Propaganda functions through the "repetition of such ideological redescriptions, values + beliefs can be codified into opinion favorable to the ideological goals of a larger strategy" (Hyzen 2023a, p. 7). The broader ideological strategy of managing opinion is larger than a single propaganda message or specific campaign and contains immediate and long-term goals, but all rely on repetition and dissemination. Propaganda is a highly flexible endeavor of manipulating "significant symbols" (Lasswell 1927, p. 627). One way to understand how the combination of manipulation of significant symbols and repetition plays a crucial role in the successful dissemination of war propaganda is through the lens of framing.

Bennet's indexing model, like Herman and Chomsky's propaganda model, emphasizes that the dominance of elite sources not only determines what media talk about but how they talk about (i.e., frame) it, by limiting "the range of voices and viewpoints in both news and editorials according to the range of views expressed in mainstream government debate" (Bennett 1990, p. 106). This reflects hegemonic notions of how these sources manage to set the frames in which news stories are discussed (Gitlin 1980). Framing is defined by Entman (1993) as "a process whereby a frame suggests which aspects of reality are selected, rejected, emphasized, or modified [providing} the audience with context and suggested meaning" (p. 52). While Entman emphasized the framing process taking place in the media, government propaganda likewise constitutes a framing process that may precede or develop alongside media framing. By choosing what to emphasize and what to omit, framing in government propaganda and media coverage influences citizens by suggesting a meaning that results from power relations.

Framing analysis is an established method in political communication (De Vreese 2005) and has been applied to propaganda (Cozma 2015; Manzoor et al. 2019; Andersen and Sandberg 2020). Qualitative framing analysis emphasizes frames as expressions of ideological processes that generalize in news themes and topics (Reese 2010). This is coherent with our conception of propaganda dissemination. Framing analysis gives insight into the essential elements of a message, reasoning devices, that are directly linked to the functions of framing by identifying the problem, causes, moral evaluation and solution or treatment (Entman 1993). Furthermore, framing well captures repetition, an essential tactic of propaganda.

Indexing, as an explanatory model for how the dissemination of state propaganda through media is possible, has been tested especially for U.S. mainstream/legacy media coverage of international affairs, conflicts and wars (Hallin 1994; Bennett 1990, 2004, 2019; Bennett et al. 2007). Entman's (2003) cascading activation model suggests that, as events such as a conflict or war progress, journalists are not passive recipients of government

propaganda, incorporating their and their audiences' interpretations into their framing. Yet, it confirms the hegemonic borders within which such "dissent" takes place, similar to Herman and Chomsky's model, and Entman's model does not undermine the idea that government sources are central to framing a conflict or crisis at its outset, the topic of this study.

This results in the final broad research question (RQ3): how do the U.S. government and U.S. media frame the Russia–Ukraine war and how are government and media framing related?

## 3. Case Set-Up: Materials and Methods

To apply the theoretical considerations to the original positioning of the U.S. towards the Russian–Ukrainian war in the early days of the conflict, we focus on the first week of the war (2/24-3/3/2022) when the U.S. executive branch and media built a distinct campaign to position the U.S., to indicate what the priorities were (agenda-setting) and how to interpret them (framing).

### 3.1. Sampling

3.1.1. Press Briefings

The positioning by the U.S. government is analyzed through the press briefings of two legs of the U.S. executive power: the White House (hereafter WH), the official representation of the President, geared at domestic audiences and international observers, and the State Department (hereafter SD), the diplomatic leg of the U.S. government. Transcripts of the six press briefings during that first week were collected: 2/24, 2/28 and 3/3 for WH, and 2/25, 2/28 and 3/3 for SD. These briefings typically start with official position statements from the respective executive branch, followed by a Q&A with journalists from the U.S. and some foreign media.

3.1.2. Media Coverage

Legacy news media, especially television news, remain crucial in war reporting and, thus, in the process of creating and disseminating propaganda. To analyze the media agenda-setting and framing and if/how they reflect government agenda-setting and framing, we focus on coverage of the war in partisan U.S. cable networks, Cable News Network (CNN), considered (at the time) to be left of center, and The Fox News Channel or Fox News, considered (conservative) right of center. CNN is a U.S.-based, multinational cable channel, created by Ted Turner and owned by Warner Bros Discovery through CNN Global. Fox News is a U.S.-based, multinational cable news television channel, created by Australian-born Rupert Murdoch and owned by Fox Corporation through Fox News Media.

The programs selected for analysis are the two cable channels' closest approximations of "regular" news programs on broadcast channels (Kim et al. 2022): *CNN Newsroom* for CNN and *Special Report with Bret Baier* for Fox News, each daily programs with a duration of 50 min (commercial breaks and announcements excluded). *CNN Newsroom* reached 696,000 households and 821,000 viewers older than two in the first quarter of 2022, the period that includes our sampled days (https://ustvdb.com/networks/cnn/shows/newsroom/). *Special Report with Bret Baier* on average had 2.87 million viewers in March 2022, then the fifth-most-watched program on Fox News (Johnson 2022).

To allow for both quantitative and qualitative analysis, for both programs, the sample was limited to the episodes of 2/24, 2/28 and 3/3. All broadcasts were provided by the Internet Archive.

### 3.2. Quantitative Coding and Analysis

Quantitative coding and analysis aimed to find what aspects of the Russia–Ukraine war were covered by the press briefings and media. To this end, all content was treated with the same coding schedule that was developed, based on the literature on war and peace journalism regarding conflict reporting (e.g., Lynch and Galtung 2010) and on the

dimensions of and indicators for propaganda in texts (verbal and visual media coverage, official statements, etc. For overviews, see Zollmann 2015, 2019), next to codes reflecting the institutional and non-institutional actors and issues potentially involved in this particular conflict. The coding schedule allowed us to indicate whether an item was about the war or not and, if so, what topic was covered: battlefield, security, peace, sanctions, war justification, U.S. policies, UN policies, NATO policies, EU policies, civilians, refugees, nuclear hazard, misinformation and other. Briefings were further coded for whether an item was part of the opening statement (i.e., agenda-setting by the government) or the Q&A, where journalists could draw attention to topics not mentioned by government officials, and whether questions received off- or on-topic responses. Briefings were coded for the number of items and media coverage for the number of items and time, as the latter is a sparse resource that affects news selection. All materials were coded by one of the authors and 10% of the coverage and briefings were coded by the other author, showing Krippendorff's alpha intercoder reliability of 0.8 or more for all nominal variables, indicating high agreement (De Swert 2012). All data were submitted to descriptive statistical analysis.

*3.3. Qualitative Framing Analysis*

The qualitative, inductive framing analysis aimed to uncover ideological messages, analyzing how war-related topics were talked about in press briefings and media. Every meaningful section of the briefings and coverage was submitted to the following questions: What/who is the problem? What/who caused it? What moral evaluation is provided What is the recommended treatment? These questions help to illuminate if and how the briefings and media coverage followed Bernays' six characteristics of war propaganda, especially in identifying the instigator and castigating blame on the enemy while justifying the home positions and interventions.

Inductive framing analysis approached the primary texts with an open mind, yet coding was informed by an understanding of general indicators of (war-related) propaganda in texts. Zollmann (2019) groups these along three dimensions: alignment and legitimizing state–corporate views and actions while omitting substantial criticism; "technical" propaganda based on distorted, manipulated facts and on criticism limited to tactics and procedures rather than substance; and demonization of actors and actions of opposing positions.

Answers for each framing question were brought together in a matrix to look for similarities, consistencies and overall patterns. These were combined into frame packages (Gamson and Modigliani 1989). Frame salience was determined by how often they occurred and relationships between frames (dominant, secondary, counter and neglected) (Zhou and Moy 2007). Special attention was paid to framing devices, including word choice, metaphors, symbols and stereotypes as part of propaganda's rhetorical toolbox, including the identification of the conflict (conflict, (un)just war, occupation or liberation, . . .), of actors involved (name-calling, glorification. . .), of events (war crimes, aggression, defense) and of events and actors (Pan and Kosicki 1993).

## 4. Results: Quantitative Analysis

*4.1. War-Related—Not War-Related News*

The six briefings had a total of 461 items, of which 60 items (13.1%) were not war-related and 401 (86.9%) were war-related. *CNN Newsroom* spent the entirety of its three analyzed programs on the war, while *Fox News Special Report* went from dedicating almost the entire news program to the war (97.19%) on the first day, 85.5% of the time on 2/28 and just 57.1% on 3/03. The remaining data analysis focuses solely on war-related reporting.

*4.2. Topics*

Quantitative analysis of the occurrence of war-related topics in briefings and news coverage suggests how the U.S. government and media tried to set the agenda regarding the war.

4.2.1. Topics in White House and State Department Briefings

First, we look at which topics were mentioned in the WH and SD briefings in that first week of the war.

Table 1 shows that more than half (54.1%) of all items in WH and SD briefings combined concerned sanctions and "other". The former dealt with the U.S.'s position towards and role in deciding on the sanctions against Russia in response to the invasion, while almost all items identified as "other" concerned the U.S. positioning itself towards the war. Closely related is the extensive attention (23.4%) given in WH briefings to U.S. policies other than sanctions. Every other topic received much less attention (<10% each). While SD briefings paid some attention to the position of the UN, this was next to absent from WH briefings, as was the position of the EU. Some attention was devoted to global security and to potential nuclear hazards either through Russian use of nuclear weapons or danger related to the Russian occupation of Chernobyl. Interestingly, the actual battlefield, Ukrainian civilians and refugees received limited attention. The WH made no mention of peace talks, and the SD somewhat (6.5%). Not surprisingly, any justification for the Russian aggression was absent.

**Table 1.** War-related topics in % of number (#) of items for White House (WH) briefings, State Department (SD) briefings and all briefings combined.

| Topics/Items % | WH ALL (#248) | SD ALL (#153) | ALL (#401) |
|:---:|:---:|:---:|:---:|
| Battlefield | 3.2 | 2.0 | 2.7 |
| Civilians | 0.0 | 2.6 | 1.0 |
| Sanctions | 34.7 | 19.6 | 28.9 |
| Refugees | 4.8 | 2.6 | 4.0 |
| Security | 7.7 | 8.5 | 8.0 |
| UN | 0.8 | 7.2 | 3.2 |
| EU policies | 0.8 | 0.0 | 0.5 |
| US/NATO policies | 23.4 | 5.9 | 16.7 |
| Justifications | 0.0 | 0.0 | 0.0 |
| Nuclear Hazard | 6.0 | 9.2 | 7.2 |
| Peace | 0.0 | 6.5 | 2.5 |
| Disinformation | 0.0 | 0.0 | 0.0 |
| Other | 18.5 | 35.9 | 25.2 |
| Total | 100 | 100 | 100 |

To better understand agenda-setting efforts by the government, we compare topics raised in the briefings' opening statements, representing the official position statement of U.S. executive powers, with the topics raised in the Q&A, where journalists could initiate other topics, suggesting similar or different areas of interest.

Table 2 shows that the opening statements firmly focused on just three topics: U.S. policies (14.3%), sanctions (28.6%) and "other" (57.1%). The latter refers, in the first WH briefing and the first two SD briefings, to extensive identification of the war as "unjustified, unprovoked, premediated attack", "Putin has made his choice", and "flagrant violation of international law", which we elaborate on in the framing analysis, and, in the second WH paper, to the position of the U.S. as the largest provider of assistance to Ukraine.

The questions from journalists followed this agenda-setting by asking further questions about sanctions (26.9% of all questions) and U.S. policies directly (16,8% of all questions identified as U.S. position) or indirectly (some of the questions coded as "other" could be interpreted as such). Yet, subsequent questions cover other issues, notably security (19% of all questions) and nuclear hazards (14% of all questions). Interestingly, 19.4% of all answers provided by officials were off-topic, with WH officials answering off-topic 17.3% of the time and SD officials 22.7% of the time. One-off topics (gathered in the category "other") were overwhelmingly introduced through questions from correspondents, were typically brief and ranged from embassies moving from Kyiv to Lviv (SD), the Kennedy Center canceling a Russian Ballet performance (WH), Russia's crackdown on independent

media (SD), providing Putin an exit strategy (WH), and reports that China asked Russia to invade after the Olympics (SD), among others.

**Table 2.** War-related topics as they occurred in opening statements, journalists' questions and officials' answers for all briefings combined.

| Topics/Opening, Q&A # % | Opening Stat. # | Opening Stat. % | Questions # | Questions % | Answers # | Answers % |
|---|---|---|---|---|---|---|
| Battlefield | 0 | 0.0 | 8 | 4.1 | 3 | 1.5 |
| Civilians | 0 | 0.0 | 1 | 0.5 | 3 | 1.5 |
| Sanctions | 2 | 28.6 | 53 | 26.9 | 61 | 31.0 |
| Refugees | 0 | 0.0 | 8 | 4.1 | 8 | 4.1 |
| Security | 0 | 0.0 | 19 | 9.6 | 13 | 6.6 |
| UN | 0 | 0.0 | 7 | 3.6 | 6 | 3.0 |
| EU policies | 0 | 0.0 | 1 | 0.5 | 1 | 0.5 |
| US/NATO policies | 1 | 14.3 | 33 | 16.8 | 33 | 16.8 |
| Justifications | 0 | 0.0 | 0 | 0.0 | 0 | 0.0 |
| Nuclear Hazard | 0 | 0.0 | 14 | 7.1 | 15 | 7.6 |
| Peace | 0 | 0.0 | 6 | 3.0 | 4 | 2.0 |
| Disinformation | 0 | 0.0 | 0 | 0.0 | 0 | 0.0 |
| Other | 4 | 57.1 | 47 | 23.9 | 50 | 25.4 |
| Total | 7 | 100 | 197 | 100 | 197 | 100 |

### 4.2.2. Topics in the Media

To compare the agenda-setting by the U.S. executive power with the agenda-setting by the U.S. media, we analyzed the occurrence of various war-related topics in the media coverage during the same opening week. As said, while for briefings, topics are measured by the number of items, media topics are presented in time, a scarce resource for news programs affecting selection and perceived importance (time spent on an item).

#### CNN Newsroom

Table 3 shows that *CNN Newsroom* had the battlefield as the most covered topic on days 1 and 3, always opening with and returning to it, followed originally by sanctions and later by civilians that dominated the second broadcast. The dominance of the battlefield (mostly absent from press briefings) follows the media logic that images of battlefields drive home the seriousness and horror of the conflict, evoking strong emotions (Chouliaraki 2013). The media logic likewise explains the considerable attention to the impact of the war on civilians, especially in the second (38.96%) and third (27.48%) selected programs, an item absent from the opening statements and nearly absent from the overall briefings. *CNN Newsroom* followed the press briefings in their attention to sanctions (and somewhat to other U.S. policies) in the first two sampled programs, to nuclear threat in the second and third programs and in their silence on the UN and EU positions, on peace and on disinformation. Like the briefings, *CNN Newsroom* provided no justification for the war. Finally, "other" topics constitute a wide range of one-off topics including Ukrainian civilian safety instructions (2/24), protest marches in various parts of Europe and the U.S. (2/28) and Putin's mental health concerns (3/3), amongst others.

#### Fox News Special Report

In Table 4, we provide a more detailed overview of the distribution of various topics across the sampled period for *Fox News Special Report*.

**Table 3.** Sample days by topics of war news in percentages of duration in *CNN Newsroom*.

| Sample Day/ Topic % | Battlefield | Civilians | Disinformation | EU | Justification | Nuclear Hazard | Peace | Refugees | Sanctions | Security | UN | US | Other | Total |
|---|---|---|---|---|---|---|---|---|---|---|---|---|---|---|
| 24 February | 65.98 | 7.68 | 0.00 | 0.00 | 0.00 | 0.00 | 0.00 | 0.00 | 14.32 | 0.00 | 0.00 | 3.49 | 8.55 | 100 |
| 28 February | 23.57 | 38.96 | 0.00 | 0.00 | 0.00 | 13.00 | 0.00 | 2.08 | 10.97 | 0.00 | 0.00 | 2.30 | 9.12 | 100 |
| 3 March | 40.53 | 27.48 | 0.00 | 0.00 | 0.00 | 12.97 | 0.00 | 6.63 | 1.41 | 0.00 | 0.00 | 0.91 | 10.07 | 100 |

**Table 4.** Sample days by topics of war-related news in percentages of duration in *Fox News Special Report*.

| Sample Day/ Topic % | Battlefield | Civilians | Disinformation | EU | Justification | Nuclear Hazard | Peace | Refugees | Sanctions | Security | UN | US | Other | Total |
|---|---|---|---|---|---|---|---|---|---|---|---|---|---|---|
| 24 February | 17.37 | 4.84 | 0.00 | 0.00 | 0.00 | 0.00 | 4.76 | 0.00 | 52.34 | 3.81 | 0.00 | 9.36 | 7.53 | 100 |
| 28 February | 21.55 | 0.00 | 0.00 | 0.00 | 0.00 | 19.78 | 0.00 | 7.37 | 27.93 | 0.00 | 0.00 | 11.53 | 11.84 | 100 |
| 3 March | 26.24 | 0.00 | 0.00 | 0.00 | 0.00 | 0.00 | 0.00 | 4.68 | 0.00 | 10.58 | 0.00 | 33.67 | 24.83 | 100 |

The breakdown of war-related topics across the sampled days shows that *Fox News Special Report*, like *CNN Newsroom*, paid considerable attention to the battlefield in each episode but that sanctions dominated the first two episodes, in that regard following the briefings more closely in the topics discussed (if not evaluation, see framing), dedicating more than half (52.34%) of its first program to it. The U.S. position was another important topic. Like *CNN Newsroom*, there was attention to nuclear hazards, both referencing potential danger from Russian soldiers occupying Chernobyl and interpretations of threatening comments by Russian president Putin. *Fox News Special Report* also followed the briefings in their silence on the UN and EU, on peace and on disinformation, and there was no justification for the war. Different from the briefings and *CNN Newsroom*, *Fox News Special Report* paid some attention to the possibility of peace talks on the day the war started. As in the case of CNN, the "other" category contains a wide range of one-off topics, including the impact of the invasion on the U.S. stock market (2/24), international sport events canceling Russian participation (2/28) and Americans going to Ukraine to help (3/3), amongst others.

## 5. Results: Qualitative Framing Analysis

The analysis of what the briefings and media talked about is complemented by a framing analysis of how these topics were talked about, starting from the briefings.

### 5.1. Dominant Frame 1: "Unjustified, Unprovoked, and Premeditated"

A first frame that dominated the WH and SD briefings and was picked up by the media coverage contextualized the Russian military invasion of Ukraine as "unjustified, unprovoked, and premeditated" and had a dominant interpretation and two subframes.

### 5.1.1. Dominant Frame 1

The U.S. government message was clear: Putin and his inner circle were firmly presented as the problem, the cause of the war and the ones to blame. The solution was opposition from the U.S. and the world community with sanctions aimed at punishing Russia, specifically Putin and his inner circle. The moral evaluation is that the invasion is unjustified on every level, therefore wrong/bad and those who oppose it are right/good. Following war propaganda strategies (cf. Bernays 1942), this frame enforces assigning war guilt (1) on Russia, to claim unity in opposition as well as "victory" with the effectiveness of sanctions (2) and to counter Russian news claiming the war was provoked, framing unfavorable information as lies (5). For example, the opening statements of the earliest WH and SD briefings have nearly identical passages: the "entire world are with the people of Ukraine today as they suffer an unjustified, unprovoked, and premeditated attack by the Russian military forces" (WH 2022a, 2/24, p. 1); and the "American people are standing with the people of Ukraine as they suffer an unjustified, unprovoked, and premeditated attack by Russia's military forces" (SD 2022a, 2/25, p. 1). On the final sample day, the SD opening statement continued the theme as "Russia's continued premeditated, unprovoked, and unjustified war against Ukraine" (SD 2022b, 3/3, p. 2). This theme was repeated as an entire phrase in the opening statement of all SD briefings but also as individual words to punctuate statements and reinforce the message, like "Ukraine in the face of Russia's unprovoked aggression" (SD 2022c, 2/28, p. 3), while the WH simplified it to "President Putin's aggression" (WH 2022b, 3/3, p. 22). *CNN Newsroom* echoed several times, "Biden condemning the attack as unprovoked and unjustified" (e.g., *CNN Newsroom* 2022a, 2/24, min 53), as did *Fox News Special Report*, notably even up to a year into the war (e.g., "U.S. determination to hold Russia accountable for crimes committed its unjust and unprovoked invasion of Ukraine"(*Fox News Special Report* 2023, 3/3b, min 44).

Both in the briefings and media coverage, this frame presents the Ukrainian people either as the suffering victims of Putin's "brutal attack" (SD, 3/25, p. 1), "a tragedy for the people of Ukraine" (WH 2022a, 2/24, p. 3), or as courageous: "the Ukrainian military, President Zelenskyy, and others, have fought courageously" (WH 2022c, 2/28, p. 10) and "the people of Ukraine continue to fight with courage and pride for their country"

(SD 2022c, 2/28, p. 3). Media coverage, especially CNN, depicted these two sides of the Ukrainian citizens in detail. This served to further the moral conclusion that Russia is bad and rallied support and unity towards Ukraine as the brave victims. In terms of war propaganda strategies, this aimed to strengthen the belief that Russia was responsible for the war (4) and to emphasize early battle victories (2).

This frame serves the propaganda strategy of the U.S. government to explain its position and actions to the American people, to serve as diplomatic effort, rallying allies' support for sanctions, and as geo-political posturing towards Russia. It justifies the U.S. position, allied with Ukraine and in opposition to Russia, with "unprovoked" attaching full responsibility to Putin and Russia and innocence to Ukraine.

The topic of sanctions dominated briefings and media and, in line with war propaganda strategies (1) and (4), was carefully framed as a response to the invasion, as severe, targeted to Putin and Russian elites, and as coordinated with allies, the latter serving the war propaganda strategy (2) by claiming unity of all allies participating in sanctions against Russia. While their policies were rarely the topic of briefings or media coverage, the EU and NATO were used as framing sponsors:

> [the] U.S. imposed an unprecedented package of financial sanctions and export restrictions in lockstep with our Allies and partners that will isolate Russia from the global financial system, shut down its access to cutting-edge technology, and undercut Putin's strategic ambitions to diversify and modernize his economy. (WH 2022a, 2/24, p. 3)

A key framing device is the repetition of rhetoric and themes, such as "the most biting, the most crippling, the most devastating set of sanctions" (SD 2022a, 2/25, p. 16), "crippling financial sanctions on President Putin, his inner circle, and the Russian economy" (WH 2022c, 2/28), "historic steps that are crippling the Russian economy", across the briefings analyzed and in many media reports. For example, Fox News reported "Western financial sanctions against [Putin's] government and his elites are putting a real strain on the Russian Economy" (*Fox News Special Report* 2022b, 2/28, min 1:01).

5.1.2. Subframe 1.1: "Putin's Choice"

An important subframe that becomes a returning theme emphasizes that the war was Putin's decision. The war blame was assigned directly to Putin as part of the war propaganda strategy (1) and to strengthen the belief of Putin's depravity (4). WH and SD briefings had congruent passages, again suggesting coordination: "Putin has made his choice. He rejected diplomacy and chose war" (WH 2022a, 2/24, p. 2) and "Putin has made a choice. He rejected diplomacy. He has chosen war" (SD 2022a, 2/25, p. 4). This theme of Putin or the Kremlin's "choice" is repeated throughout the briefings, for example, to punctuate a sentence "because of Putin's choices" (WH 2022a, 2/24, p. 4) or to contextualize: "this isn't the first time, of course, Putin has decided that his country can attack another country with impunity" (SD 2022a, 2/25, p. 3). CNN's WH correspondent picked up this characterization, referring to "Vladmir Putin's assault on Ukraine" (*CNN Newsroom* 2022a, 2/24, min 7:34). It was also used to differentiate Putin and Russian leadership from Russian citizens: "a choice Putin made . . . for the people of Russia. This is a choice that has been decided for them, not by them" (SD 2022a, 2/25, p. 4). Continuing, *CNN Newsroom* quoted the 2/24 WH press briefing referring to Putin and his war, reading Secretary Psaki's statement "despite the Kremlin's propaganda, there are Russian people who profoundly disagree with what he is doing in Ukraine" (*CNN Newsroom* 2022a, 2/24, min 19:30). Putin's choice was also used to support sanctions: "inflation is skyrocketing, the Ruble is the worst-performing currency in the world. It was his decision to go to war" (WH 2022a, 2/24, p. 27). It was also used in conjunction with war propaganda strategy (5) to claim the enemy's information is false: Putin was "falsely alleging that it is Russia that is under threat [. . .] that Russia was the one that had no choice" (SD 2022c, 2/28, p. 8).

"Putin's war of choice" is a significant phrase and propaganda theme, appearing repeatedly in multiple briefings (WH 2022a, 2/24; WH 2022b, 3/3; SD 2022a, 2/25; SD

2022b, 3/3) and media coverage. On the first day of the invasion, *CNN Newsroom* ran a tinker headline quoting the Biden administration that read "Putin chose this war" (*CNN Newsroom* 2022a, 2/24, min 38). In fact, *CNN Newsroom* continued to carry the phrase well into the war, framing a report on day 51 of the war as "Putin's war of choice" (*CNN Newsroom* 2022f, 4/14, min 1), confirming the media's lasting adoption of official U.S. verbiage.

### 5.1.3. Subframe 1.2: "Putin as Irrational and Unstable"

This subframe relates to the trope of the paranoid, evil emperor, especially the Russian dictator Joseph Stalin. This frame was introduced by the media, not U.S. officials, in questions during briefings and segments of analyzed media coverage by *CNN Newsroom* and *Fox News Special Report*. However, during a briefing, U.S. officials confirmed their previous statements referring to Putin's then-recent speeches as increasingly "bizarre" (SD 2022c, 2/28, p. 16). Framing devices introduced in questions during briefings included "We see him issuing nuclear threats. So I wonder, do you still consider him a rational actor?" (SD 2022c, 2/28, p. 16). Putin's mental fitness was discussed to contextualize his nuclear threats in both SD and WH 2/28 briefings, with officials repeatedly commenting that Putin's remarks were escalating or escalatory. Both programs seized on the term the same day, *Fox News Special Report* with a segment graphic titled "It is Escalatory" (2/28, min 4:46) and *CNN Newsroom* reporting that Putin's nuclear comments were a "further escalation" (2/28, min 9:27). *CNN Newsroom* and *Fox News Special Report* broadcast stories that directly mentioned or questioned Putin's mental health or capacities. A *CNN Newsroom* story reported that the French President Macron claimed Putin was an isolated and changed man since they last spoke, while the written headline stated "growing questions about Putin's grasp on reality" (*CNN Newsroom* 2022b, 2/28, min 7:04–10:26; ticker headline, 8:45). Mikhail Khodorkovsky, a former oligarch who spent 10 years in a Russian prison, claimed Putin was extremely cruel and shows clinical paranoia (*CNN Newsroom* 2022c, 3/3, min 44:28–48:31), while Dominic Thomas was interviewed, agreeing with former President Obama's statement that Putin is not the same person, irrational (*CNN Newsroom* 2022e, 4/7, min 6:53–12:49). *Fox News Special Report* ran a story specifically questioning Putin's mental state and capacity, claiming he was unhinged, that former U.S. Secretary of State Condoleezza Rice and others felt he became a madman, noting that Putin does not use a cellphone or the Internet (*Fox News Special Report* 2022b, 2/28 min 45:20–47:23). Another *Fox News Special Report* headlined that Putin was visibly upset, lashing out at top military commanders for their failures (*Fox News Special Report* 2022d, 3/24, min 22:17–24:33).

### 5.2. Dominant Frame 2: "America, Defender of the Free World"

A second dominant frame in the briefings, picked up by the media, contextualizes American involvement in the Russia–Ukraine conflict and its siding with Ukraine: "the United States has undertaken to rally the world for democracy and against Russian aggression" (WH 2022c, 2/28, p. 21) and had one subframe.

### 5.2.1. Dominant Frame 2

This frame is based on wider socio-political references that unmistakably evoke Cold War era propaganda of the "Free-World" vs. the U.S.S.R. Statements from WH and SD fitting this frame represent continued coordinated messaging for domestic and international consumption. The frame identifies Russian invasion and coercion as the problem, while U.S. aid and leadership are the solution. The moral evaluation is that Russia, as the aggressor, represents the bad, i.e., totalitarianism, injustice and corruption, while the U.S. represents the good, i.e., freedom and democracy, justice and rule of law. Moreover, Ukrainians deserve U.S. protection as they, too, fought for their freedom, "as the people of Ukraine continue to fight with courage and pride for their country, we will continue to provide them the assistance that they need" (SD 2022c, 2/28, p. 3). This framing mainly follows the war propaganda strategies to claim unity (2) and to represent and project international law, security, social order and peace (3) but also to place war guilt on Russia (1) and strengthen

the belief that Russia is both corrupt and responsible (4). Notably, U.S. involvement was never fundamentally questioned in the briefings (nor media), creating a crucial premise to the frame: that the U.S., as a leader and protector of the free world, was involved and would be an active political participant in this war. Following this premise, U.S. government messaging followed two tracks: the U.S. showed leadership by providing the most aid and military assistance to Ukraine and the U.S. was leading unified allies against tyranny and injustice.

Regarding U.S. aid to Ukraine, the "head of USAID, Samantha Power, is on the ground in Poland and has been there as well, providing guidance" (WH 2022c, 2/28, p. 2) and the U.S. would support "their security needs, their humanitarian needs, their economic needs, their political needs" (SD 2022a, 2/25, p. 19). U.S. commitment was resolute: "The United States is the largest provider of assistance to Ukraine" (WH 2022c, 2/28, p. 1), including a "package of up to \$350 million for immediate support to Ukraine's defenses, bringing the total security assistance over the past year to more than \$1 billion in support of Ukraine's frontline defenders" (SD 2022c, 2/28, p. 3) and the U.S. was "expediting military defense assistance" (WH 2022b, 3/3, p. 16). In all briefings, aid and support to Ukraine was mentioned and emphasized. *CNN Newsroom* broadcast statements from Zelensky calling for aid (*CNN Newsroom*, 2/24 min, 37:04) and a member of the Ukrainian parliament calling for more weapons (*CNN Newsroom* 2022a, 2/24, min 24:26). Likewise, *Fox News Special Report* interviewed the Ukrainian Foreign minister who called for weapons and lethal aid (*Fox News Special Report* 2022a, 2/24 min 10:30). Both *CNN Newsroom* and *Fox News Special Report* covered U.S. officials on aid and calls for aid from Zelensky (*CNN Newsroom* 2022b, 2/28, min 47:05) and *Fox News Special Report* aired segments explaining battlefield weapons, training and what America could supply (*Fox News Special Report* 2022c, 3/3a, min 31:55).

Regarding the U.S. leading unified allies towards certain victory (2), hedging against disunity and defeat (5), framing devices included that this was "about the unity of the vast majority of the global community in standing up against President Putin, it's more about American leadership in this moment, and it's more about even unity here in standing up against the aggression of President Putin" (WH 2022c, 2/28, p. 24). While the U.S. represents the "Free-World", "Putin's assault on Ukraine is an attack on the principles that undergird global peace, stability, and security the world over" (SD 2022a, 2/25 p. 2). Putin was seen as "one of the greatest unifiers of NATO in modern history [...] you see a NATO Alliance that was incredibly unified and agreement among Europeans that there needed to be additional assistance provided in the form of security assistance to Ukraine" (WH 2/28, p. 3). Rhetorical repetition underlined this: "the response to Russia's war has been unity–unity among world leaders, unity in Europe, unity among people gathering around the world to protest President Putin's war of choice, including thousands of people in Russia and Belarus" (SD 2022b, 3/3, p. 3). The media referenced this, too, for instance when *CNN Newsroom* stated "Zelensky says Russia has not just started against Ukraine but also the entire democratic world" (*CNN Newsroom* 2022a, 2/24, min, 52). Another tactic suggested disunity between Putin and citizens, claiming the Russian populace were "rejecting [Putin's] violence against the people of Ukraine" and were "calling on their government to stop this unprovoked and unjustified war" (SD 2022a, 2/25, pp. 3–4). *CNN Newsroom* reported the Russian government cracking down on domestic war protesters and making arrests (1742 people) (*CNN Newsroom* 2022a, 2/24, min 18:27–19:47).

### 5.2.2. Subframe 2.1: Russia Violates the Rules and Norms of International Relations

A subframe emphasizes that Russia and Putin broadly violated international norms and rules, disrupted serious international diplomacy and engaged in illegal activities. These themes relate to propaganda strategies to place war guilt on Russia (1), to strengthen the beliefs that Putin and the Russian elite were responsible (2), to position the U.S. as representing law and order, peace and stability (3), and to counter Russian news and information that contradicted the U.S. perspective (4). It also led to the branding of this war as illegal, another propaganda technique (Legal Filter) used to demonize Russia

while insinuating the U.S. represents justice, legality and "the Free World" (Hyzen 2023b). Russia and Putin were seen to violate the "international rules-based system" which was intertwined with rhetoric and language of criminal activity: "Putin's war of choice has required that we follow through on imposing the massive consequences and severe costs, and that we ensure his flagrant violation of international law will be a strategic failure" (SD 2022a, 2/25, p. 4). The framing as criminality includes phrases such as "Russian elites who are complicit in Putin's kleptocracy and their family members" (WH 2022a, 2/24, p. 2) and "we are going to hunt down and freeze the assets of Russian companies and oligarchs [and] any other ill-gotten gains that we can find and freeze under the law" (SD 2022c, 2/28, p. 20). This gives sanctions, as a form of warfare, the aura of a "just" response, in accordance with the law, to unjust/illegal aggression, keeping open the possibility of a provoked war which is legal if a state has no choice. As a proxy war, the U.S. is within its right to retaliate with harsh sanctions.

Despite this self-ascribed leadership and moral high ground, peace talks were notably absent from U.S. official statements (cf. above). Answering a journalist's question "President [Biden] said that he thinks that Putin is go and try to expand back the Soviet Union", U.S. officials responded, "President Putin has more—has grander ambitions in Ukraine. Hence, the military campaign is continuing" (WH 2022a, 2/24, p. 33). This is a framing device of Cold War era propaganda of the free world vs. the U.S.S.R. U.S. officials furthermore insinuated that diplomatic talks were deceptive and a tactic to continue military operations, claiming: "Moscow was using the so-called pretense of diplomacy, pretending to take part in serious diplomatic discussions" (SD 2022a, 2/25, p. 15). The messaging asserted that the United States, allies in Europe, OSCE and the NATO–Russia Council engaged in good-faith diplomatic efforts to avoid war but "at every turn, the Russian Federation rejected those offers of substantive, constructive engagement" (SD 2022c, 2/28, p. 10). U.S. officials claimed they were "never going to take diplomacy ever off the table. But again, now is not the moment for that" (WH 2022b, 3/3, p. 7), yet the repeated message was that "diplomacy at the barrel of a gun, diplomacy at the turret of a tank–that is not real diplomacy" (SD 2022c, 2/28, p. 10).

*5.3. Counter-Frames*

There are no counter-frames questioning the enemy or invasion. Fundamentally, there was agreement between the media and government that the problem was Russia, and especially Putin. Counter-frames question the position of the U.S. on sanctions, energy policy and diplomacy.

5.3.1. Counter-Frame 1: "Sanctions: Too Weak, Too Clumsy?"

A counter-frame to the dominant framing of sanctions was identified in the press questions during the briefings and was implicit in the government's defense of those U.S. sanctions. The counter-frame contests the U.S. officials' framing of the listed sanctions as the best solution following two tracks, one suggesting that sanctions are too weak or slow to be effective and another that sanctions could hurt Western economies or the Russian people (rather than elites), by not sanctioning Russia's energy sector. *Fox News Special Report's* WH correspondent espoused this counter-frame on the first day of the invasion, producing an entire segment questioning sanctions' effectiveness, concluding that sanctions were a weak response from the U.S. and from Biden specifically (*Fox News Special Report* 2022a, 2/24 min, 12:37).

This frame is implicitly rebuked by U.S. officials' claims that their sanctions were principled and designed "to be responsible, to avoid even the perception of targeting the average Russian civilian and, of course, unwanted spillovers back to the U.S. or the global economy" (WH 2022a, 2/24, p. 5) and that the U.S. would contain economic ripple effects by coordinating "in lockstep with our Allies and partners. And because we think the spillover effects will be manageable" (WH 2022a, 2/24, p. 11).

The government framing of sanctions was further questioned by a journalist: "So the world just sits back and watches that happen until these sanctions take effect?" (WH 2022a, 2/24, p. 8) and "Is it fair to say, then, in the immediate term the U.S. is basically unable to stop Putin?" (SD 2022a, 2/25, p. 11). U.S. officials responded by deflecting to long-term policy: "Europeans need to do that; we need to do that. If we do more to invest in clean energy, more to invest in other sources of energy" (WH 2022b, 3/3, p. 10). In an expert interview, *CNN Newsroom* reported that the EU was sending a lifeline to Russia by buying energy. Germany was trying to move towards renewable sources; however, its policy was at odds with Russian sanctions (*CNN Newsroom* 2022e, 4/7, 6:53–12:49).

5.3.2. Counter-Frame 2: "It's Biden's Unjustified War on Carbon"

*Fox News Special Report,* while covering the Russia–Ukraine war, regularly questioned and criticized Biden's energy policy. Discussions about energy-related sanctions were reframed from Russia waging war against Ukraine to Biden waging war against oil. The most notable segment was a guest appearance by former *Special Report* host Brit Hume. Hume introduced what he called Biden's "War on Carbon", strongly advocating to repeal environment protections, instead building nuclear power plants, taking the "shackles" off domestic energy corporations, framing this as the only way to hurt Russia and Putin with sanctions (*Fox News Special Report* 2022b, 2/28, min 13:11–16:53). In reporting on a speech in which Biden called Putin a Pariah and villain, *Fox News Special Report* dug up a video statement from Biden at a 2021 diplomatic summit where he identified Putin as "bright", "tough" and a "worthy adversary" (*Fox News Special Report* 2022b, 2/28 min, 13:50), suggesting inconsistency. These types of edits appeared across different reports from anchors, reporters, interviewees and panels, reflecting Fox's domestic political and propaganda goals.

*CNN Newsroom* ran a critical segment questioning why the U.S. was not targeting Russia's energy sector for sanctions (*CNN Newsroom* 2022d, 3/7, min 20). Though specific media were not identified in briefing transcripts, reporters questioned U.S. officials on energy policy and sanctions exempting Russian energy sales, to which officials replied, "this is the one area where Russia has systemic importance in the global economy [...] That's not to say that we have a dependence on Russia; Russia depends on those revenues just as much as the world needs its energy" (WH 2022a, 2/24, p. 7).

**6. Discussion**

Taking the first week of the Russian–Ukraine war, we combined a quantitative topic analysis and qualitative framing analysis of U.S. official press briefings and U.S. partisan cable network news to understand key characteristics of propaganda dissemination in times of war, crucially focusing on state–media relationships.

We found that U.S. officials connected with lingering Cold War propaganda narratives to characterize the Russia–Ukraine conflict as Russia vs. the "Free World", implying Putin's goals beyond Ukraine were akin to the re-establishment of the U.S.S.R. U.S. officials presented a professional, disciplined and WH-SD-coordinated propaganda campaign to set the agenda and frame the war and, thus, dictate the government's desired narratives to the media by focusing on certain topics and specific interpretations or frames. As such, we confirmed that U.S. messaging complied with the six principles of war propaganda (Bernays 1942), predominantly by using true information. By maintaining a tightly controlled script with returning phrases and themes, like "unprovoked, unjustified and illegal" or "Putin's war of choice", U.S. officials aimed to downplay competing narratives and unfavorable stories and, in some cases, ignore some details altogether.

Both left-of-center *CNN Newsroom* and right-of-center *Fox News Special Report* followed the government logic and fundamental premises, including not questioning U.S. involvement (as the government ensured "no boots on the ground" or direct air support) and the Cold War framing of Russia and Putin as totalitarian aggressors against Ukraine as brave victims representing the West and democracy and, therefore, worthy of defense by the

leader of the free world. This fits the Herman and Chomsky (1988) propaganda model. Likewise, *CNN Newsroom* and *Fox News Special Report* adopted the phrase "Putin's war" or "Putin's aggression" in reports along with personal criticisms of Putin. Along Cold War lines, *Fox News Special Report* in particular depicted China and Russia as close allies and enemies of the U.S., thus against freedom and democracy.

Both media outlets spent much of the broadcast time on battlefield and airstrike reports, conforming to standard media logic for war reporting. While this attention to the battlefield did not occur in U.S. official statements, it served the crucial propaganda strategy of strengthening the belief that the war is the enemy's fault (4) and supported this with horror stories from the conflict (6). *Fox News Special Report* coverage focused on battlefield and airstrike reports, nearly always broadcast with footage of battles and destruction. *CNN Newsroom*, part of a media brand that was built on war reporting, strongly emphasized war stories, tragedy, interviews with fleeing refugees and sheltering civilians, wounded soldiers and civilians, and graphic footage.

The case confirms that propaganda originates from both the government and media in interesting ways. The U.S. government succeeded to a significant extent in getting specific messages and fundamental contexts of the Russian invasion across to and repeated in the media, confirming Herman and Chomsky and wider propaganda model studies as well as the indexing model that put journalist–elite source relations at the heart of this process. At the same time, the media also provided their own focus, closely related to media logic, especially in the attention to (strong images of) the battlefield. The partisan media we covered also introduced their own frames, some that partly countered (aspects of) the state's position. Notably, at nearly any opportunity, *Fox News Special Report* heavily criticized the Biden government and Democratic politicians like Nancy Pelosi on energy policy. However, they were never in disagreement about the core of the U.S. position, confirming the hegemonic work of indexing. Interestingly, *Fox News Special Report* relied heavily on the Defense Department's briefing and quoted the briefing on the first day of the war, avoiding the Biden administration's politicized communications, perhaps showing their partisanship.

Crucially, our analysis showed how propaganda was generated mostly through facts rather than mis/disinformation. Barring (future) discrediting of events aside, we found largely accurate media accounts of U.S. sanctions, airstrikes, destruction, the Moskva sinking, and the number of Russian soldiers crossing the borders or Ukrainian refugees fleeing. Official statements from Putin, Biden and Zelensky were not false but rather carefully curated by the media.

At the same time, important information was absent from both the government briefings and media coverage. There was very little mention of peace by U.S. officials, who deflected the topic with statements like "diplomacy is highly unlikely to bear fruit, to prove effective, in the midst of not only confrontation but escalation" (SD 2022c, 2/28, p, 10), nor was it covered by the U.S. media, whereas media in other countries devoted significant coverage to it (Nordenstreng et al. 2023). This reflects that peace and deconfliction were of low strategic priority in communications. Likewise, briefings nor media offered information regarding Ukraine's tenuous history with democracy and corruption (Kuzio 2015), rather framing Ukraine as a victim desperately trying to retain its free, liberal democracy. This was an immediate change in rhetoric from previous skepticism of governments and media towards Ukraine joining institutions like the EU or NATO, related to Ukraine's historical inclusion as a Cold War enemy. Critically, the government and media demonstrated a lack of contextualization of the current conflict in the Russian annexation of Crimea in 2014, let alone Russia's attack on Georgia in 2008. Geopolitical drivers of the conflict, arguably a proxy war between world powers, and the recent history of the region were absent from briefings and media coverage. Above all else, this absence helps in achieving the U.S.'s central propaganda goal: not controlling every frame or issue but dominating the premises on which the issues are discussed.

### 7. Conclusions

This contribution provided a comparative quantitative and qualitative analysis of official U.S. communications and U.S. partisan media coverage in the first week of the Russian invasion of Ukraine. It aimed to uncover (RQ1) how the U.S. government propagated the Russia–Ukraine war and its involvement in it; (RQ2) to what extent the U.S. media picked up U.S. government propaganda attempts to set the agenda; and (RQ3) how the U.S. government and partisan media framed the Russia–Ukraine war and how the government and media frames were related. Theoretically, we combined propaganda studies with indexing theory to understand the working of propaganda in the early days of the conflict with a focus on state–media relations. The results suggest, regarding RQ1, a narrow focus and distinct framing of the conflict on the part of the U.S. government. Regarding RQ2, we show that partisan media partly took over the government agenda-setting, while also touching on other topics that fit media logic, especially relating to the battlefield. Likewise, regarding RQ3, the results show that, in terms of framing, partisan media provided some counter-frames in line with their ideological positions, yet overall conformed to the dominant U.S. government framing of the war as unjust, unprovoked and premeditated, as Putin's choice, and the position of the U.S. as the leader of the free world and defender of democracy.

Our study confirms and contributes to Zollmann's (2019) call to demarginalize propaganda studies by (re)integrating them into journalism and political communication studies. We show propaganda to be as much about (reconfigured) facts and information as about disinformation and lies and, in the case of state propaganda, to be closely related to indexing (cf. Kennis 2009). Based on journalist–elite source relations, it results in the media following elites' agenda-setting and framing of events, particularly in the early stages of this event. Our study further highlights that legacy media remain crucial in state propaganda efforts, and, thus, are a worthy focus of analysis, the existence of propaganda notwithstanding. Methodologically, our study shows the importance of combining quantitative and qualitative approaches. Especially in a time that favors big data analysis, our study shows the relevance of a qualitative, inductive approach in unearthing the intricacies of propaganda.

The combined quantitative and qualitative approach does come with the limitation of a small data sample that covers only a short period of state and media communication and that is restricted to the executive branch of government and television. As such, our study findings regarding how U.S. state propaganda helped to set the agenda and framing of the Ukraine–Russia war coverage in the U.S. cannot be extended beyond the highly charged starting period of the conflict or extrapolated to other media. For instance, longer-term tracking of propaganda may show evidence of Entman's (2003) "cascading network activation" model that suggests that over time, political and other elites, media and audiences contribute to evolving news frames, in line with hegemonic room for discussion and disagreement as Herman and Chomsky's propaganda model acknowledges. Entman's model also reminds us to consider the (active) audience, something our study does not touch upon and that certainly needs attention in follow-up research. However, the absence of an audience and a longer-term perspective does not undermine our key findings regarding the strength of the relationship between executive power and media in the original agenda-setting and framing at the outset of a crucial conflict.

**Author Contributions:** Conceptualization, A.H. and H.V.d.B.; methodology, A.H. and H.V.d.B.; formal analysis, A.H.; investigation, A.H. and H.V.d.B.; resources, A.H.; data curation, A.H.; writing—original draft preparation, A.H. and H.V.d.B.; writing—review and editing, A.H. and H.V.d.B. All authors have read and agreed to the published version of the manuscript.

**Funding:** This research received no external funding.

**Institutional Review Board Statement:** Not applicable.

**Informed Consent Statement:** Not applicable.

**Data Availability Statement:** Data for the qualitative framing analysis are publicly available. White House Press Briefings can be accessed here: https://www.whitehouse.gov/briefing-room/press-briefings/ (accessed on 9 January 2023). The State Department briefings can be accessed here: https://www.state.gov/department-press-briefings/ (accessed on 9 January2023). The CNN and Fox News footage can be obtained from Internet Archive: Digital Library of Free & Borrowable Books, Movies, Music & Wayback Machine (accessed on 11 July 2022).

**Acknowledgments:** We want to thank R. Macdonald and K. Leetaru at the Internet Archive for helpful support in obtaining the video materials; D. Yanich and H. Wolgast at the University of Delaware for their input in the development of our coding schedule; Em. K. Nordenstreng (Tampere University) and the team of international colleagues involved in analyzing early reporting of the Ukraine-Russia conflict (Nordenstreng 2023); J. Van den Bulck (University of Michigan) and K. Panis (Thomas More University of Applied Sciences) for their helpful advice on the quantitative analysis.

**Conflicts of Interest:** The authors declare no conflicts of interest.

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
