# Peer review of "“Putin’s War of Choice”: U.S. Propaganda and the Russia–Ukraine Invasion"

_journalmedia, doi:10.3390/journalmedia5010016_

Round 1

Reviewer 1 Report

Comments and Suggestions for Authors

This is an interesting and original study of US propaganda and selected news media coverage of Russia's 2022 invasion of the Ukraine. The study is based on a solid theoretical discussion of propaganda and news media. This is used as a backdrop for an original primary data analysis of US government press briefings as well as news media reporting by selected broadcasters during the first week of Russia's invasion. The article would benefit from the following revisions:

Comments on the theoretical part:
- The theoretical section on Propaganda and War (2.1.) is useful. However, the author suggests that propaganda is about realizing ideological goals which are then defined as values, beliefs and opinions. Yet, the author then argues that propaganda is conceptualised as a value-neutral phenomenon (p.3). This is unclear as the examples given by Lasswell and Bernays (which appear to be confirmed later in the empirical study) are not value neutral. Perhaps the author could further clarify these issues to obtain better theoretical coherence.

- The section on state propaganda and the media leaves out some important literature. The author refers to an "usually uneasy cooperation between the executive branch and the fourth estate of the media" and suggests that the US government has been successful in managing the media (p. 4). How this process of media management works in terms of the relationships between government and media is only insufficiently explored. There are economic as well as organisational reasons and also sourcing relationships that explain why media rely on government propaganda. This literature could be further explored and added. This would help explaining why the US's press briefings that are later analysed in the study might have had an impact on the media. In this context, the author mentions Herman and Chomsky's propaganda model arguing it would predict collusion between government and media. But how this works is not explained. There are in fact processes that are embedded in media structures (like use of official sources) and which do not necessarily constitute collusion. Such issues should be further unpacked with more nuance.

There are some methodological issues:

- The empirical study of CNN and Fox News only assesses three episodes per sender during week one of the invasion (p. 6). This is a small sample and impacts on the generalisability of the findings. Ideally, the sample should be expanded. If this is not possible it needs to be better explained why such a small sample was selected and what the limitations of the design are. For example, it could be argued that this is an explorative study that combined looking at government briefings and news media based on qualitative and quantitative analyses and such a broad study design only allowed to look at a small sample.

- There is not sufficient alignment between theoretical concepts of propaganda and how propaganda was detected in media texts. A theoretical elaboration of content indicators for propaganda studies (which have been defined by other scholars) might be helpful. The study refers to themes by Lasswell and Bernays which are very interesting. But these need to be translated into content dimensions as well as indicators that can be identified in media texts. Also, the study compares how topics like battlefield, civilians, sanctions etc. were covered in US government press briefings as well as media. However, the connection between how the press briefings and the media handled these topics is vague. So, the author should more clearly map out how the statements from the press briefings link to the media discourse. This was partly done in the qualitative analyses. However, there appears to be more discussion of press briefings as opposed to the media discourse. It would have been better, for example, to investigate if the chosen news media reported statements from the designated press briefings. Such a study could better demonstrate if the media directly picked up statements from the briefings rather than just including similar themes that could have been part of the media narrative for a variety of reasons. Additionally, the dominant narratives from the press briefings could have been unpacked and linked to more examples from media coverage. So, overall, the quantitative and qualitative media analyses need to be more coherently linked with content indicators for propaganda. More examples from the media should be provided/analysed and more coherence between press briefings and media content could be established.

Other points:
- There are some inconsistencies. The author writes on page 2: "The opening quote does not necessarily hold untruths: many Americans stand with the Ukraine, the war is arguably unprovoked, does not follow the rules of a just war, and the invasion must have been planned. Yet, the message is highly charged, ideological propaganda."

It is unclear how this message can be highly charged, ideological and not holding untruths at the same time.

Additionally, many scholars and commentators have argued that the war was provoked by NATO expansion although it is an illegal war. So, propaganda might involve some facts that are true but it is not based solely on true facts. This quotation is an example of that as the US government avoids talking about their own role in the war in terms of provoking Russia. Such complexities should be considered. This means that propaganda, while it may contain truthful elements, has to be theorised as a mix of distortions and truths. If a statement does not include untruths, as the authors suggest in this case, it is not propaganda but something else.

- It is not clear how the concept of "hot" and "banal" propaganda adds to the argument. This should be further explained or the concept could be left out.

- The author uses three RQs but these RQs seem not to be addressed in the conclusion/discussion. It would be good to pick these up in the final section and summarise main findings per RQs.

- It is not appropriate to have one final discussion/conclusion section. The study should have a discussion section that links important findings back to the theory/content indicators/literature and then afterwards a final conclusion section.

- The primary sources and their bibliographical information should be in a separate reference list as well.

Comments on the Quality of English Language

Overall fine, some minor revisions to enhance clarity might be necessary.

Reviewer 2 Report

Comments and Suggestions for Authors

In quantitative analysis, when the category of "other" is more than 10%, it suggests the need for recoding. It is also not clear how exactly coding was done: how many coders participated, what was the intercoder reliability, etc.

I would avoid using the word "slogan" for "themes." In propaganda studies, the word "slogan" has a very specific meaning that does not correspond to the analysis in this project. Also, I would avoid assessment phrases such as "to demonize Russia" (line 594) and "proxy war" unless these phrases are the actual part of the official documents analyzed in the study. Also, subframe "Putin is crazy" shows bias, and from the analysis it was not clear if it is the researchers' bias or the journalists'. I would replace it with "Putin is not a rational actor" or something to that effect.

I do not see enough evidence that the US government statements revived the Cold War rhetoric. Also, the word "banal" (line 680) seems to be out of place in this context. When Russia actively positions itself as engaged in the proxy war with the entire West and NATO, perhaps the first step to the revival of the Cold War rhetoric was made by Russia and the US simply had to respond in-kind?

In general, it seems that the authors have a negative bias toward the US communication messages that needs to be toned down for the academic article.

Comments on the Quality of English Language

Overall, the language is good, although there were minor mistakes such as "Ukrainians deserves" (line 537), "the entire the world" (line 27), "where" instead of "were" at the end of line 654, "suggestion" instead of "suggesting" in line 663 and the use of "the" before "Ukraine" (line 78) that may be seen as the anti-Ukrainian bias.

Round 2

Reviewer 1 Report

Comments and Suggestions for Authors

The authors did address my revision suggestions with good attention to detail. This is now an interesting and academically sound study and I recommend publication.

There are some small inconsistencies in the writing (U.S. at times written as US etc.) and I would recommend a further proof reading of the article to catch such minor writing errors.